# Insights into the Structure–Capacity of Food Antioxidant Compounds Assessed Using Coulometry

**DOI:** 10.3390/antiox12111963

**Published:** 2023-11-03

**Authors:** Francesco Siano, Anna Sofia Sammarco, Olga Fierro, Domenico Castaldo, Tonino Caruso, Gianluca Picariello, Ermanno Vasca

**Affiliations:** 1Istituto di Scienze dell’Alimentazione, Consiglio Nazionale delle Ricerche, Via Roma 64, 83100 Avellino, Italy; francesco.siano@isa.cnr.it (F.S.); olga.fierro@isa.cnr.it (O.F.); 2Dipartimento di Chimica e Biologia “A. Zambelli”, Università degli Studi di Salerno, Via Giovanni Paolo II 132, 84084 Fisciano, Italy; anna3sofia@gmail.com (A.S.S.); tcaruso@unisa.it (T.C.); 3Stazione Sperimentale per le Industrie delle Essenze e dei Derivati dagli Agrumi (SSEA), Azienda Speciale CCIAA di Reggio Calabria, Via G. Tommasini 2, 89125 Reggio Calabria, Italy; dcastaldo@ssea.it; 4Ministero delle Imprese e del Made in Italy, Via Molise 2, 00187 Roma, Italy; 5Dipartimento di Ingegneria Industriale e ProdAl Scarl, Università degli Studi di Salerno, Via Giovanni Paolo II 132, 84084 Fisciano, Italy

**Keywords:** antioxidant capacity, coulometry, electrogenerated bromine, biamperometric detection, CDAC, antioxidant mechanisms, electrochemical ratio

## Abstract

CDAC (coulometrically determined antioxidant capacity) involves the determination of the antioxidant capacity of individual compounds or their mixtures using constant-current coulometry, with electrogenerated Br_2_ as the titrant, and biamperometric detection of the endpoint via Br_2_ excess. CDAC is an accurate, sensitive, rapid, and cheap measurement of the mol electrons (mol e^−^) transferred in a redox process. In this study, the CDAC of 48 individual antioxidants commonly found in foods has been determined. The molar ratio CDAC (CDACχ, mol e^−^ mol^−1^) of representative antioxidants is ranked as follows: tannic acid > malvidin-3-*O*-glucoside ≃ curcumin > quercetin > catechin ≃ ellagic acid > gallic acid > tyrosol > BHT ≃ hydroxytyrosol > chlorogenic acid ≃ ascorbic acid ≃ Trolox^®^. In many cases, the CDACχ ranking of the flavonoids did not comply with the structural motifs that promote electron or hydrogen atom transfers, known as the Bors criteria. As an accurate esteem of the stoichiometric coefficients for reactions of antioxidants with Br_2_, the CDACχ provides insights into the structure–activity relationships underlying (electro)chemical reactions. The electrochemical ratio (ER), defined as the antioxidant capacity of individual compounds relative to ascorbic acid, represents a dimensionless nutritional index that can be used to estimate the antioxidant power of any foods on an additive basis.

## 1. Introduction

Redox homeostasis is an essential function that regulates the dynamic balance between reducing and oxidant compounds in living organisms and modulates the physiological response to many biological processes that occur in an oxygen-rich environment. In the human body, the disruption of redox homeostasis and the subsequent imbalanced presence of oxidizing species can injure cellular and tissue structures and trigger aging-related damage and degenerative pathologies, including metabolic syndrome, rheumatoid arthritis, atherosclerosis, hypertension, cardiovascular diseases, and cancer [1,2]. While low amounts of reactive oxygen (ROS) and reactive nitrogenous species (RNS) may have physiological regulatory and defense functions, cells and biological tissues are endowed with a complex arsenal of compounds capable of inactivating the excess of harmful species produced under conditions of oxidative stress. Human endogenous antioxidants encompass a range of protein and enzymatic modulators alongside several low-molecular-weight metabolites [3,4].

Diet affects the redox balance of the human organism to a large extent. Sound epidemiological evidence associates well-being and health-promoting effects with the intake of various classes of antioxidant compounds, which are particularly abundant in fruits and vegetables [5,6]. Antioxidants play important technological roles for food preservation as well, by inhibiting or delaying the oxidative degradation of their biomolecular constituents [7]. For instance, food antioxidants can act at varying levels as quenchers for the singlet O_2_ trigger or chain-breaking inhibitors of lipid peroxidation. Given their nutritional and technological relevance, antioxidants are among the most extensively studied classes of chemical species. According to a commonly accepted comprehensive definition, an “antioxidant is any substance that delays, prevents or removes oxidative damage to a target molecule” [8]. Although often used interchangeably, “antioxidant activity” and “antioxidant capacity” must be interpreted differently. Antioxidant capacity refers to the amount (expressed in moles) of a free radical that can be scavenged by an antioxidant [9,10]. Antioxidant activity is a kinetic parameter and refers to the reaction rate between an antioxidant and an oxidant species. In this sense, the antioxidant activity is related to the conditions of the assays with which it is evaluated, while the concept of antioxidant species might lose any biological significance outside an experimental environment [11].

In general, foods, plant extracts, and biological fluids feature heterogeneous assortments of antioxidant classes of compounds. What is improperly referred to as “total antioxidant capacity (TAC)” is a measurable parameter that should account for the comprehensive antioxidant contribution of the various species in a given matrix [12]. An extraordinarily large number of chemical assays have been proposed for the determination of the antioxidant potential of both individual compounds and synergistic phytochemical mixtures in many different matrices [13]. Most of these assays measure the radical scavenging capacity or the reducing ability, which is different from the preventative antioxidant capacity of a compound or mixtures of compounds. The thermodynamic and kinetic aspects of the mechanisms involved are complex and often undefined, undermining the accuracy and significance of the measurements. Diverse assays correlate poorly with each other since they are sensitive to specific classes of compounds, while excluding several others [6,14]. Thus, some of the major challenges in antioxidant testing are to establish which method is best suited to any application and which could be the physiological relevance of the in vitro assessment of the antioxidant capacity. Related to this latter aspect, the evaluation of the in vivo effects of exogenous antioxidant compounds should reckon with “pharmacokinetic” parameters such as bioaccessibility, ADME (absorption, distribution, metabolism, excretion), and capability to interact with target biomolecules [9].

Due to the inherent drawbacks that each of them suffers from, none of the antioxidant capacity assays can find universal applicability. Antioxidant capacity is measured in an extraordinarily number of investigations concerning foods and phytochemical extracts, with practically no standardization of both experimental procedures and expression of results. Similarly, despite the health relevance of the biomarkers associated with redox status, no standard clinical diagnostic analysis based on the evaluation of the TAC in biological fluids has been developed so far [15,16]. Attempts to classify the antioxidant capacity of phenolic compounds have been performed in silico using quantum chemical approaches [17]. However, these models have scarce correlations with experimental approaches and their reliability is limited to simplistic chemical conditions different from those found in foods or in the human body.

Based on the pioneering work of Uchiyama and Muto [18], Abdullin et al. [19] proposed the use of electrogenerated Br_2_ as a universal titrant for the coulometric determination of the antioxidant capacity of phytocomplexes. Electrogenerated Br_2_ (standard potential E^0^ = +1.05 V) oxidizes rapidly and quantitatively most of the known naturally occurring antioxidants (e.g., phenolics, ascorbic acid, terpenes, carotenoids, tocopherols, and their derivatives). The stoichiometry of the redox processes is hardly predictable, but—aside from a few exceptions—the number of electrons that is transferred during the conversion of antioxidants, calculated using the Faraday equation, is dependent on the number of reactive functional groups in the structure [20]. Mainly because of its sensitivity, constant-current coulometry with electrogenerated Br_2_ has been successfully exploited to assay the phenolic compounds excreted in human urine [21] or to assess slight variations in the antioxidant capacity in human blood [22]. The micromoles of electrons delivered for oxidizing the target analyte(s) are measured with high accuracy when compared to conventional volumetric redox titration. The so-called coulometrically determined antioxidant capacity (CDAC), which has been applied to the evaluation of the antioxidant capacity of extra virgin olive oil micronutrients [23,24] and *Citrus* spp. agrifood byproducts [25], represents an intriguing application of conventional coulometric titration. The CDAC measurements substantially consist of a constant-current coulometric titration, in which the titrating agent is in situ electrogenerated Br_2_. A schematic of the device developed and the details for measuring CDAC have been previously reported [23]. Briefly, the coulometric circuit can be represented as follows:(+) Pt |TS| |1 M KBr| Pt (−)
where TS is the test sample.

The micromoles of the electrons delivered (µ) in the circuit through which the constant current I (mA) flows for t seconds can be easily calculated using the Faraday equation:µ = I × t/96.487(1)

For the sake of dimensional consistency, in (1), the Faraday constant (96,487 C (mol e^−^)**^−^**^1^) is divided by 1000.

While a constant current is flowing in the circuit (I), Br_2_ is produced via the reaction:2Br^−^ → Br_2_ + 2e^−^(2)

As far as oxidizable species occur in the solution, they are oxidized by Br_2_ with reactions and mechanisms that remain substantially undefined, while each bromine atom is formally reduced into Br^−^. When oxidizable species are completely consumed, the simultaneous presence of Br_2_ and Br^−^ in the solution generates an increasing current *i* (μA) in a biamperometric circuit held at a constant potential. Excess Br_2_ is stepwise generated in the solution until a current significantly different from the noise is detected. The end-point is determined on the *x*-axis (μmol e^−^) from a *i* vs. μ plot via linear extrapolation of no less than four *i* vs. t data points. In this way, a limitation of the approach suggested by Ziyatdinova et al. [26], which is markedly dependent on the accuracy of a single amperometric current measurement, can be overcome.

At the endpoint of each titration, the CDAC value (μmol e^−^ g^−1^) is calculated using the following equation:CDAC = (μ − μ_0_)**/**(V × C)(3)
where μ_0_ indicates the micromoles e^−^ delivered before the addition of the samples so that the electrogenerated Br_2_ oxidizes any pre-existing interfering compounds (blank measurement), and V is the volume (mL) of the solution of each standard at the concentration C (mg mL^−1^). The quantities μ and μ_0_ are readily obtained from the *i* vs. t plots.

As for the other constant-current coulometric determinations, CDAC does not require the use of standards, and boasts several other operative advantages over the most common assays for antioxidant capacity [20], especially including sensitivity, accuracy, time-/cost-effectiveness, low environmental impact, and readability of the results. Thanks to the accurate determination of the micromoles of electrons exchanged, the method is suitable for measuring the antioxidant capacity of poorly water-soluble compounds. For the above reasons, CDAC could be a candidate reference method for the assessment of the in vitro antioxidant capacity of individual compounds, their complex mixtures, or biological fluids, and could be suited for the standardization of procedures and expression of results. In this work, the CDAC values of a selection of 48 pure antioxidant compounds, including some naturally found in foods and other commonly used as preservatives, were determined to rank their antioxidant potency and to establish structure–antioxidant capacity relationships.

## 2. Materials and Methods

### 2.1. Chemicals

Representative antioxidant compounds were selected, among many other possible ones, based on their distribution in foods and easy availability. The 3,4-dihydroxy phenylacetic acid was purchased from Alfa Aesar (Ward Hill, MA, USA), while all the other standards, of analytical grade and of the highest purity available, were from Sigma (St. Louis, MO, USA). The hydroxytyrosol was synthesized by reducing 3,4-dihydroxyphenylacetic acid with LiAlH_4_ in dry THF under refluxing, according to a previously reported procedure [27]. The reaction was monitored using silica gel thin-layer chromatography (TLC) with ethyl acetate (EtOAc):petroleum ether (EP) 6:4 (*v*/*v*) as the eluent and stopped after 2.5 h. The reaction mixture was cooled and, after work-up with EtOAc and water, the organic layer was concentrated under reduced pressure. The resulting oil residue was dissolved in EP/EtOAc under heat and the EP proportion was increased, cooling the mixture to promote crystallization. Afterward, the solid precipitate was checked using analytical TLC and high-pressure liquid chromatography with diode array detection (HPLC-DAD) under the previously described conditions [24], confirming that it was the expected hydroxytyrosol in comparison with a polar extract from extra virgin olive oil. The purity of the hydroxytyrosol was >90% as assessed using HPLC analysis.

### 2.2. CDAC Measurement

A made-on-purpose prototype electrochemical device integrating both the coulometric circuit and the biamperometric detection system (CDAC 2.0, by Microbees srl, Naples, Italy) was used to determine the antioxidant capacity via coulometry.

In the coulometric circuit, a platinum ring (Metrohm, Oreggio, VA, Italy, mod. 6.0351.100) was the anode, while the cathode, hosted in a separate glass tube ending with a glass frit (G3 porosity), was a platinum wire (Metrohm, mod. 6.0301.100). The biamperometric detector consisted of a double platinum sheet electrode (0.15 × 8 × 8 mm, Metrohm, mod. 6.0309.100). The determinations were performed in a 150 mL flat-bottom, closed glass vessel with four 14/23 normalized ground glass necks, three of which served for placing the electrodes and the latter for sample introduction. The glass vessel contained a 1 M KBr solution; in the case of acidic analytes, it also contained 0.1 M H_2_SO_4_. Each CDAC measurement was performed with 0.1 mL of a methanol solution of individual molecules, accurately prepared at a known concentration. After each determination, the Pt electrodes were cleaned with 1/1 (*v*/*v*) 65% HNO_3_/H_2_O and repeatedly rinsed with double-distilled water. The CDAC values were measured in triplicate at least and averaged.

## 3. Results

The mechanisms through which antioxidants exert their activity can substantially be classified into the following three categories [28]:(i)Electron transfer (ET) reactions from the antioxidant to the substrate(ii)Hydrogen atom transfer (HAT) reactions from the antioxidant to the substrate, which, in aqueous media, can be considered as a proton transfer combined with an electron transfer; it can occur in one step or involve mechanistically distinct ET and proton transfer steps(iii)Chelation of metal ions by antioxidants, which inhibits the genesis of free radicals.


ET and HAT mechanisms are not exclusive and can operate simultaneously, as in the case of phenolic compounds. Thus, a given antioxidant can participate either via one of three mechanisms or via mixed mechanisms depending on the compound structure, chemical environment, pH, and nature of the free radical. Antioxidant capacity assays exploit one of these mechanisms and, as a coulometric method, CDAC should be classified within the assays based on ET. The CDAC of a species or a mixture of compounds provides an index of the antioxidant capacity because the electrochemical oxidation could be intended as a measure of the radical scavenging capacity [29].

The electrogeneration of Br_2_ involves the production of other bromine species in minor amounts, such as Br_3_^−^ and short-living Br·. Altogether, these species react quickly according to multiple reaction mechanisms (radical, redox, electrophilic substitution, and addition to multiple bonds) to convert oxidizable compounds [19]. Simultaneously, a part of the generic electroactive species “a” can be converted from the reduced (Red_a_) into the oxidized (Ox_a_) form directly at the platinum anode, according to the following reaction:Red_a_ → Ox_a_ + ne^−^


The remaining Red_a_, which generally is the largely prevalent part, is oxidized by the in situ produced Br_2_. The overall determination of delivered electrons accounts for all these possible side events involving electron transfer, in addition to the mere redox exchange [30]. On the other hand, no electroactive reactants or products that may bias the measurements are adsorbed on the electrodes [31].

### 3.1. CDAC of Ascorbic Acid and Definition of the Electrochemical Ratio (ER)

The CDAC values of 48 pure substances commonly recognized as antioxidants and naturally occurring in foods or used as food preservatives were measured and are reported in Table 1. These substances were grouped according to their chemical classes. Both weight (mmol e^−^ g^−1^ or µmol e^−^ mg^−1^) and molar ratio CDAC values (CDACχ, mol e^−^ mol^−1^) are reported for each species assayed in this study. The validation parameters, accuracy, and precision assessment of the CDAC assay have been reported previously [23]. Practically, CDACχ provides the number of electrons participating in the reaction of the antioxidants with electrogenerated Br_2_, which indicates the free radical scavenging capability [32].

Br_2_ converts ascorbic acid into dehydroascorbic acid, which has no residual antioxidant properties [33]. The stoichiometry of the process involves the transfer of an electron pair that finally coincides with the net release of two hydrogen atoms.

The value of CDACχ = 2.0 clearly reflects the transfer of 2 moles of electrons per mole of ascorbic acid and the absence of other side processes that may distort the CDAC values from the theoretical stoichiometry. Notably, the value 2 coincides with the stoichiometric coefficient determined for the oxidation of ascorbic acid induced by the vanadium (V) → vanadium (IV) reduction [19], considering that the standard reducing potential of the vanadium (V)/vanadium (IV) couple is very close to that of the Br_2_/Br^−^ couple.

A typical plot for CDAC determination, comparing the individual measurements of ascorbic acid and gallic acid, is shown in Figure 1. For equimolar amounts of substances, the distance between µ values, extrapolated via interception on the *x*-axis, provides a visual comparison of the antioxidant capacity of a generic substance with that of ascorbic acid.

Because of such a well-defined and stable CDAC, as well as its diffusion and importance in the context of food micronutrients, ascorbic acid was selected as the reference compound to determine the electrochemical ratio (ER) of individual compounds, which is therefore defined as the antioxidant capacity of a species relative to ascorbic acid, expressed with a dimensionless index as a ratio between the mmol e^−^ g^−1^ values (Table 1). The ER intends to provide an intelligible index of the antioxidant power of an individual species. More interestingly, the ER could allow estimating roughly, on an additive basis, a theoretical antioxidant capacity of any food with a known composition. In other terms, the ER might provide a valuable nutritional index to be used for food labeling, as previously proposed but not yet made available due to the lack of standardized methods for estimating antioxidant capacity [15]. With a similar purpose, Carlsen et al. [34] launched the Antioxidant Food Database, which collects the antioxidant capacity of pure antioxidants, spices, herbs, fruits, and plant-derived foods, determined using a modified ferric reducing ability of plasma (FRAP) test. A more recent open-source comprehensive antioxidant database stores details about 56 thousand small molecules and some thousands of peptides and proteins tested for their antioxidant capacity using varying assays [35].

The CDACχ of Trolox^®^ (CDACχ = 1.8), which is a water-soluble analogue of tocopherol often used as a reference to express the relative antioxidant capacity, was very close to that of ascorbic acid, so that the ER can be considered a rough estimate of Trolox^®^’s equivalent capacity too.

### 3.2. Structure–Antioxidant Capacity Relationship 

Among the individual compounds tested in this study (Table 1), the greatest CDACχ value (CDACχ = 58) was recorded for tannic acid. Despite often being described with a defined molecular formula corresponding to decagalloyl glucose, the commercial form of tannic acid consists of a mixture of polygalloyl glucoses or polygalloyl quinic acid esters with galloyl moieties varying in the 2–12 range depending on the plant source. Moreover, the reaction between tannic acid and electrogenerated Br_2_ is complex. For this reason, the number of Br_2_ moles consumed in the coulometric titrations per 1 mole of tannic acid can vary in a wide range [32].

Flavonoids comprise a large class of plant metabolites, such as flavanones, flavones, isoflavones, flavonols, anthocyanins, and proanthocyanidins, all characterized by a C6-C3-C6 skeleton, with two benzene rings (A and B) linked by a heterocyclic pyrane ring (C) (Figure 2).

The substitution of the C-ring defines the class of flavonoids, while the substituents on the A- and B-rings define the members of each class. Flavonoids are biosynthesized via the phenylpropanoid pathway starting from shikimate and passing through the conversion of phenylalanine into *p*-coumaroyl-CoA, which represents the common precursor of the pathway branches leading to hydroxycinnamic acids, lignans, coumarins, chalcones, and stilbenoids as well. Most of the health-promoting properties of flavonoids must be ascribed to their excellent antioxidant capacity [36]. The phenolic hydroxyl group (-OH) can undergo pH-dependent deprotonation and the resulting phenolate anion can inactivate free radicals via fast electron transfer (sequential proton loss electron transfer mechanism). However, significant differences exist within this class of compounds depending on the structure, concentration, and chemical environment, as also emerging from the current coulometric data (Table 1).

Anthocyanidins feature the typical flavylium cation (2-phenyl-1-benzopyrilium) structure that actually occurs only in acidic conditions (pH < 2). pH variations greatly affect the charge, electronic distribution, geometrical conformation, and shape of anthocyanidins, modulating their reactivity and many of their functional properties. In general, anthocyanidins and anthocyanins (*O*-glycosilated anthocyanidins) exert powerful antioxidant effects, combining metal ion chelating action and free radical scavenging activity, acting as donors of either a H-atom or single electron [37]. Consistently, the anthocyanin malvidin-3-*O*-galactoside (CDACχ = 16.8) exhibited particularly high CDAC values. The CDACχ of the anthocyanidin pelargonidine (aglycone) (CDACχ = 9.2) was significantly lower than malvidin-3-*O*-galactoside.

As for anthocyanins, the oxidation of flavonoids involves the participation of the aromatic hydroxyl groups. The structural traits that concur to determine the antioxidant capacity of flavonoid congeners are described as the Bors criteria (Figure 2): (i) the presence of o-dihydroxy groups in the B-ring, particularly relevant to their chelating aptitude; (ii) the 2,3-double bond establishing the conjugation between the A- and B-rings; and (iii) the 3- and 5-hydroxyl groups in the C- and A-rings, together with a 4-oxo function in the C-ring [38]. The flavonol quercetin (CDACχ = 9.7) meets all these conditions and is commonly described as a powerful antioxidant [16]. Nevertheless, the ranking of the antioxidant capacity does not always comply with the Bors criteria. Indeed, kaempferol, which shares with quercetin the 3-flavonol backbone, but differs in the missing 3-hydroxyl group in the B-ring that abolishes the catechol function (i.e., the *o*-dihydroxy groups in the phenol rings), exhibited a significantly higher antioxidant capacity (CDACχ = 14.3) than quercetin, in line with previous assessments [39]. The flavone apigenin (5,7,4′-trihydroxyflavone) had an even higher antioxidant capacity (CDACχ = 17.2), although it does not meet two out of three Bors criteria. It should be emphasized that the order of antioxidant capacity can vary with the evaluation methods and, therefore, it should be not surprising that in some specific conditions, the Bors criteria are not respected. For instance, the chelating capacity of flavonoids featuring the catechol moiety (Bors 1 criterium) cannot be appreciated in the current experimental conditions in which transition metal ions are lacking. In this sense, the ranking of CDACχ values shows some correlation with the ORAC (oxygen radical absorbance capacity) HAT-based determinations and poor agreement with ABTS and DPPH radical scavenging assays [40].

**Figure 2 antioxidants-12-01963-f002:**
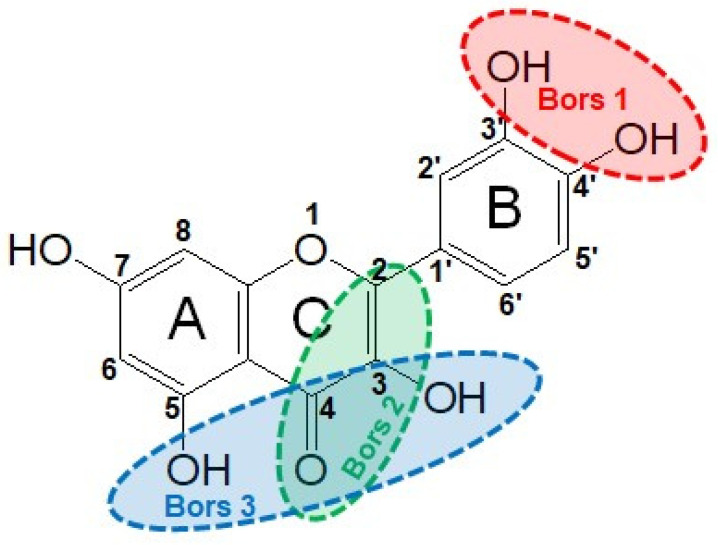
Functional groups responsible for the antioxidant capacity of flavonoids according to the Bors criteria. Bors 1: catechol moiety on the B-ring; Bors 2: 2,3 double bond and 4-oxo group on the C-ring; Bors 3: OH groups at position 3 and 5 OH group on the A- and C-rings and 4-oxo group on the C-ring. Numbers label the atom position within the flavonoid structure. The figure has been adapted from [15,40].

Catechin (CDACχ = 8.7), a flavan-3-ol lacking the 4-oxo function and the 2,3-double bond in the C-ring with the loss of both inter-ring conjugation and geometrical planarity, exhibited an only slightly lower CDACχ value compared to quercetin. This finding suggests that in this case, the o-dihydroxy groups in the B-ring might afford a dominant contribution to the antioxidant capacity.

In addition to the chelating property, the intramolecular hydrogen bond formed by the hydroxyl groups of catechol moiety promotes H-atom transfer, increasing the π-delocalization in the B-ring and enhancing the radical scavenging aptitude [40]. This should be also the case for hydroxytyrosol (CDACχ = 2.3) and its acidic derivative, i.e., 3,4-dihydroxyphenylacitic acid (CDACχ = 2.0). However, the CDACχ values of these compounds were lower than the one recorded for tyrosol (CDACχ = 3.3), despite a bulk of evidence demonstrating that hydroxytyrosol is a much more efficient peroxyl scavenger than tyrosol. The CDACχ values for 3,4-dihydroxyphenylacitic acid and hydroxytyrosol are in line with those of *o*-diphenols that oxidize to generate *o*-quinones, with no additive antioxidant mechanisms operating. A recent review summarizes the studies on the antioxidant capacity of olive oil phenolic compounds, emphasizing that many other factors besides the hydrogen atom donation capability concur to the antioxidant capacity [41]. Anyway, the clear higher reactivity toward oxidants and the more powerful antioxidant capacity of hydroxytyrosol compared to tyrosol remain to be fully elucidated on a mechanistic basis.

Oleuropein (glycosylated form, CDACχ = 3.8), which is a hydroxytyrosol derivative, exhibits a relatively low antioxidant capacity also in other radical scavenging assays such as DPPH, likely due to steric hindrance, the non-planar conformation of the molecule, and some selectivity of the kind of radicals involved [41,42]. As a confirmation of the importance of HAT mechanisms, coumarin, which lacks labile H-atoms, was unreactive with Br_2_ in our conditions. Coumarins have been described as powerful radical scavengers, but the mechanism of the antioxidant activity of non-phenolic derivatives is atypical and remains undisclosed [43]. On the other hand, it has been reported that coumarin absorbs one molecule of bromine to yield a dibromide derivative that promptly eliminates hydrogen bromide to form bromocoumarin [44]. Probably, the bromine addition to coumarin does not occur in the absence of an opportune catalyzer.

The oxidation of hydroxycinnamic acid derivatives, such as chlorogenic (CDACχ = 2.1), caffeic acid (CDACχ = 3.6), *p*-coumaric acid (CDACχ = 3.9), ferulic acid (CDACχ = 2.9), and cynarine (CDACχ = 3.6), induces a transfer of electrons ranging from 2 to 4, which underlies the formation of stable *o*-quinones and their further partial evolution into dimeric di-*o*-quinones [45]. Slight differences between CDACχ values might be also related to possible secondary interactions (e.g., hydrogen bonds) between the functional groups of the analytes or their reaction products [46].

Simple phenolic compounds such as vanillic (CDACχ = 0.8), butylated hydroxytoluene (BHT, CDACχ = 2.4), and syringic acids (CDACχ = 1.4) exhibited relatively low CDAC values. In contrast, gallic acid can undergo oxidative dimerization into dehydrodigallic acid and subsequent oxidation into the corresponding di-*o*-quinone, or, alternatively, the unstable *o*-quinone of gallic acid can undergo pH-dependent condensation and dimerization.

The semiquinone radical of gallic acid produced by oxidative electron transfer can be stabilized by resonance. The resonance structures vary depending on whether the oxidative electron transfer occurred from the *meta*- or *para*-OH group (Figure 3). Radical chain termination can occur via the coupling of any two radicals. In the specific case of the coupling of two radicals generated by an electron transfer from the *meta*-OH group, in an acidic solution, a downstream intramolecular esterification can lead to ellagic acid (Figure 3) [47]. Galloquinones can undergo coupling as well to form rather unstable dimers that may further evolve [48]. The multiple and sequential reaction pathways justify the relatively high number of electrons transferred upon the reaction of gallic acid with Br_2_ (CDACχ = 3.7). Similarly, ellagic acid (CDACχ = 8.8) is oxidized to the corresponding di-*o*-quinone and might dimerize [49]. However, the catechol-like moiety of ellagic acid can be responsible for multiple free radical scavenging processes based on sequential HAT mechanisms, which may induce additional electron transfer events in the reaction with Br_2_ [50]. The occurrence of side reactions and downstream processes likely involving the consumption of Br_2_ might justify the non-integer stoichiometric coefficients recorded for the reactions of many species.

The high number of electrons transferred in the reaction of Br_2_ with piceide (CDACχ = 12.5), which is *O*-glycosylated resveratrol (3,5,4′-trihydroxystilbene-3-*O*-β-D-glucopyranoside), implies the ability of resveratrol to trap free radicals and its tendency to form variously combined dimers and further cyclization products via oxidative radical coupling [51].

Remarkably, all these observations demonstrated that the determination of CDACχ can provide valuable clues on the chemical mechanisms of the oxidation processes.

As a general feature, the glycoside moieties are inert to the titrant and are expected to contribute to the CDAC at a minor amount [32]. Accordingly, quercetin and rutin exhibited comparable values (mCDAD = 9.7 and 11.6, respectively). On the other hand, an exceptionally high antioxidant capacity was recorded for the flavanone aglycone hesperetin (CDACχ = 19.3), even significantly higher than its glycosides hesperidin and neohesperidin (CDACχ = 11.6 and 13.4, respectively), differing from each other only by the sugar moieties.

The higher antioxidant capacity of hesperetin compared to its glycosylated derivatives has been already documented and recently confirmed using several assays [40,52]. The radical scavenging potential of flavanones is reduced by the *O*-glycosylation at the C-7 position due to steric effects that hinder the electron delocalization [53]. However, glycosylation affects the antioxidant capacity in a complex manner, and the effects change with the antioxidant capacity assay [54], so that the glycoside/aglycone pairs should be evaluated individually [55]. Interestingly, the in vivo produced metabolites of hesperitin are much more powerful antioxidants than the unmodified compound, probably because of improved water solubility [56].

The CDACχ of curcumin (CDACχ = 16.2) was relatively high, in line with its remarkable in vitro antioxidant properties. Curcumin exerts radical scavenging activity via electron delocalization involving both the phenol moiety and the methylene group within the β-diketone function or via H-atom extraction from either of these two sites [34]. Furthermore, curcumin is relatively unstable under various chemical conditions and can decompose into feruloylmethane and ferulic acid, which are electroactive species and can proceed the electron transfer process [57].

Glutathione and several representative amino acids were tested for their CDAC. Glutathione is a tripeptide considered one of the most powerful endogenous antioxidant compounds. Because of the simultaneous exchange of two electrons and two protons from two molecules of GSH with its disulfide form (GSSG), the GSSG/2GSH pair functions as a cellular redox buffer.

Cysteine (CDACχ = 5.8) and reduced glutathione (GSH, CDACχ = 4.9) exhibited a relatively high antioxidant capacity, which depends on the presence of a free sulfhydryl function and on the free electron pair of sulfur. The thiol group of cysteine (pKa = 8.45) is almost completely protonated at a physiological pH. The deprotonation of a thiol group would lead to the formation of a highly reactive thiolate anion (RS^−^), which can trigger a variety of different oxidative modifications. However, the sulfhydryl function can undergo sequential oxidation via one- or two-electron mechanisms [58]. In the presence of Br_2_, S-containing amino acids/oligopeptides react with the titrant with the formation of disulfides, which are further oxidized into sulfonic and sulfoxide derivatives [22]. Methionine (CDACχ = 1.9) has a significantly lower antioxidant capacity because the thioether lacks the possibility of a proton transfer. However, in an O_2_-rich environment, methionine can be progressively oxidized into sulfoxide and sulphone derivatives, justifying the CDACχ values comparable to ascorbic acid.

Ergothioneine is a naturally occurring amino acid found in quite high amounts in actinomycetes, cyanobacteria, and some fungi, also including edible ones. From a structural standpoint, it is the thiourea derivative of hercynine that is the betaine of histidine and contains a sulfur atom bonded to the 2-position of the imidazole ring. In solution, ergothioneine occurs in the thiol (pK_a_ = 10.8) and thione tautomeric forms. Its strong antioxidant capacity was confirmed with a CDACχ = 9.9, which is practically double that of glutathione. Ergothioneine is a powerful scavenger of hydroxyl radicals (OH∙), acting in both electron and hydrogen atom transfer mechanisms. Furthermore, it prevents the hydroxyl radical (OH∙) formation of chelating iron or copper ions to impede the catalyzed homolytic decomposition of hydrogen peroxide (H_2_O_2_). The oxidation of ergothioneine leads to disulfide and sulfonic acid derivatives [59]. In vivo, ergothioneine may serve as a non-toxic buffering antioxidant, and it may find applications in food or pharmacological preparation to prevent oxidation [60,61].

Histidine is commonly described as an effective singlet oxygen scavenger and an antioxidant amino acid [62]. However, in the current conditions, histidine was inert to the reaction with Br_2_, most likely because the pH-dependent electron donor capability is inactivated by the protonation of the weak basic imadazole group at a slightly acidic pH (pKa histidine = 6.04). It is not possible to carry out a coulometric analysis at an alkaline pH to measure the CDACχ of deprotonated histidine since OH^−^ ion is reactive with Br_2_. Hercynine, which shares the imidazole group with histidine, did not react with Br_2_ as well, demonstrating that the thiourea function of ergothioneine is necessary to the radical scavenging activity.

α-lipoic acid (CDACχ = 3.5), also named thioctic acid, exhibited a quite high Br_2_-reducing capacity due to the presence of two sulfur atoms. The reduced form of α-lipoic acid, namely dihydrolipoic acid, which was not tested in this study, is expected to have an even higher antioxidant capacity, as well as transition metal chelation properties exerted by two free thiol groups. The α-lipoic/dihydrolipoic acid pair can exert antioxidant effects via the quenching of free radicals in biological systems, protecting multiple biomolecular structures owing to its solubility in both aqueous and hydrophobic media. The oxidized form (α-lipoic acid) can be recycled in living cells [63].

In line with previous results, caffeine, furfural, and vitamin B_2_ (riboflavin) did not react with electrogenerated Br_2_ and did not contribute to CDAC [19,47].

The antioxidant capacity of essential oils and therein terpenes has been already determined via constant-current coulometry using electrogenerated Br_2_ [20]. In agreement with these findings, the CDACχ values of limonene (CDACχ = 3.5) and linalool (CDACχ = 4.2) were consistent with the bromination of the double bonds in their structure. The value determined for ergosterol (CDACχ = 1.6) was compatible with the addition of Br_2_ to one double bond.

In line with previous results [10], sulfite ions and SO_2_ (CDACχ = 0.6), which are often used as preservatives in foods and beverages, were found scarcely reactive to Br_2_. As concerns other common antioxidants used as food preservatives, sorbate exhibited a relatively high antioxidant capacity (CDACχ = 12.0), resulting from the addition of bromine to conjugated double bonds that can occur via different pathways up to the formation of tetrabromo derivatives, while butylated hydroxytoluene (BHT) had CDAC values consistent with those of simple phenols (CDACχ = 2.4).

## 4. Conclusions

Antioxidant capacity emerges as a prominent parameter of food quality and functionality, and it is frequently determined to assess the potential effects of food components and human health. The assessment of the redox status of a food matrix could be a useful chemical indicator of food quality and stability, fruit ripeness, or post-harvest storage effects. Similarly, the redox status in cells, biological fluids, and tissues has long been investigated as a diagnostic or prognostic index of the pathological events triggered by oxidative injury in humans.

The indiscriminate use of antioxidant capacity has been rightly criticized since the health effects of antioxidants are not trivially correlated with their intake or with their concentration in foods or body fluids [64]. On the other hand, the epidemiological evidence that associates a reduced risk of aging-related diseases with regular consumption of antioxidant-rich foods is undisputed.

Beyond the chemical and biological meaning, the antioxidant capacity assessment and expression imply a further layer of complexity. As a coulometric assay, CDAC is a sensitive, cheap, rapid, and reliable method to assess the antioxidant capacity of compounds or their complex mixtures. Interestingly, coulometric methods do not depend on the capability of target antioxidants to interact with specific probes as in DPPH, ABTS, FRAP, ORAC, and similar assays. Instead, they determine the intrinsic capability of a chemical species to participate in transfer processes of electrons or hydrogen atoms, often followed by downstream or side reactions, which underlie an antioxidant’s power. In this work, the antioxidant capacity of many common antioxidant compounds has been determined using CDAC to establish and classify their intrinsic antioxidant potential. The comparative study led to the following CDAC ranking of antioxidant capacity: tannic acid > malvidin-3-*O*-galactoside ≃ curcumin > quercetin > catechin ≃ ellagic acid > gallic acid > tyrosol > BHT ≃ hydroxytyrosol > chlorogenic acid ≃ ascorbic acid ≃ Trolox^®^, in substantial agreement with the data obtained using the classical FRAP assay [10]. Notably, hesperetin, apigenin, and kaempferol exhibited a higher antioxidant capacity than quercetin, contravening the Bors criteria. Among the sulfurated compounds assayed in this study, ergothioneine showed the highest antioxidant capacity.

In general, the discrepancy between CDAC and the expected stoichiometric coefficients of reactions with Br_2_ proves the subsistence of downstream or side events in addition to the primary reaction, suggesting that antioxidant mechanisms can be much more complex than a simple electron of proton transfer. The mol e^−^ transferred in the reactions with electrogenerated Br_2_ provides useful insight into the complex transformations that antioxidants undergo in the redox processes ruling the chemical aspects of antioxidant capacity. In this regard, the present study will be extended to the integration of the CDAC values with the results of voltametric experiments to be performed on the same substrates.

The ER (mmol e^−^ g^−1^), here defined as the ratio between the CDAC of individual compounds and ascorbic acid selected as the reference antioxidant, is proposed as a practical index to categorize substances based on their antioxidant capacity and to calculate on a mere additive basis the “potential” antioxidant capacity of a food matrix.

## Figures and Tables

**Figure 1 antioxidants-12-01963-f001:**
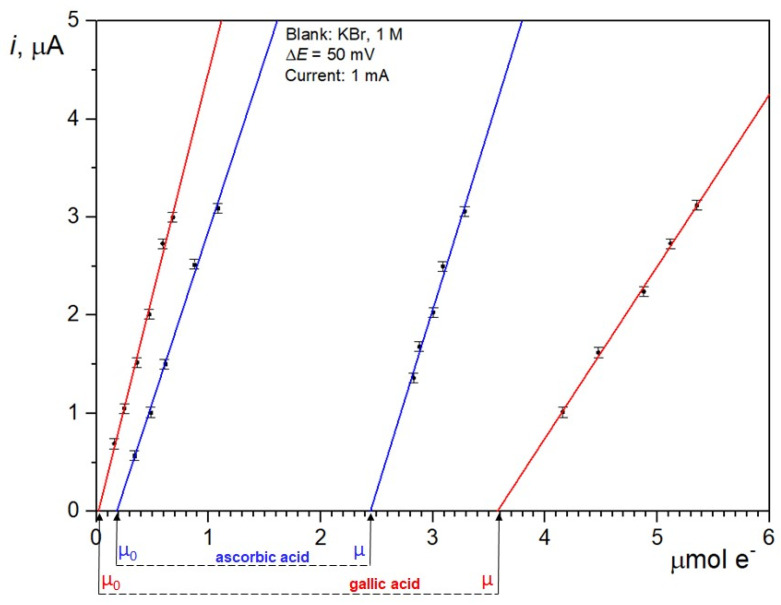
Example of plot used to determine CDAC. (μ − μ_0_) values are evaluated for ascorbic acid (blue lines) and gallic acid (red lines). Error bars represent the uncertainty of the biamperometric detection system, which is in the order of 0.05 μA.

**Figure 3 antioxidants-12-01963-f003:**
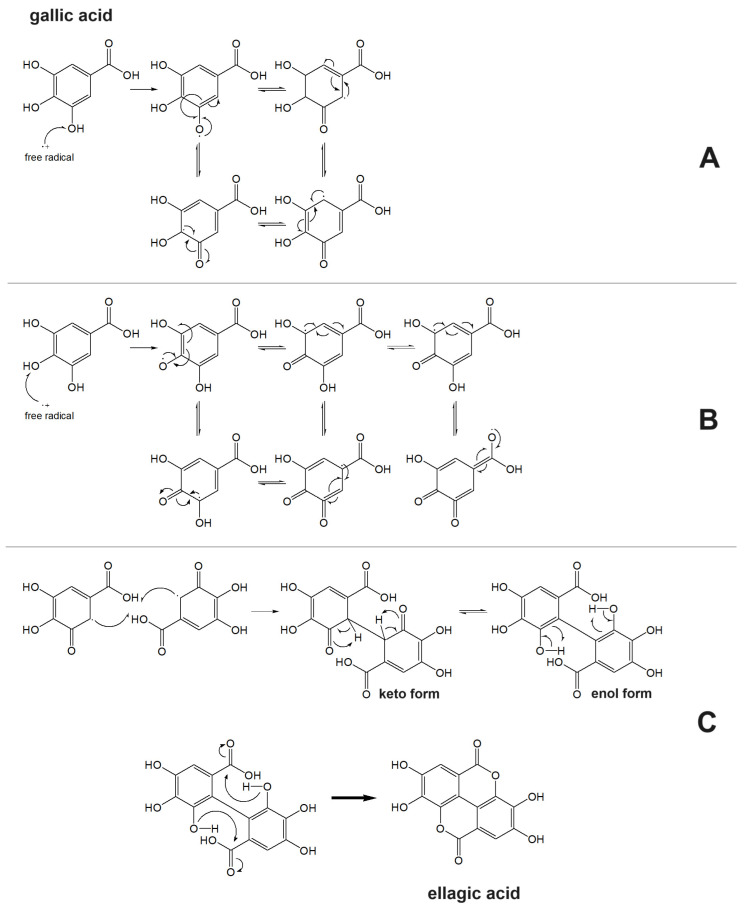
Resonance stabilization of gallic acid radical generated by the electron transfer from the *meta*- (**A**) and *para*- (**B**) OH groups. (**C**) Radical chain termination via coupling of semiquinone radicals and formation of ellagic acid following intramolecular esterification [47]. Single-barbed arrows indicate single electron shifts; double-barbed arrows indicate electron pair shifts.

**Table 1 antioxidants-12-01963-t001:** Determination of CDAC, CDACχ, and ER values for 48 pure compounds. The coefficient of variability of measurements, evaluated as relative standard deviation, was in all cases < 2% and it has been omitted.

Compound	Structural Formula	CDAC(mmol e^−^ g^−1^)	MW(g mol^−1^)	CDACχ(mol e^−^ mol^−1^)	ER	Major FoodSource
**Alkaloids**						
*Caffeine*	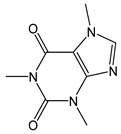	-	194.20	-	-	Coffee, tea,cocoa, guarana
**Amino acid and** **derivatives**						
*Cysteine*	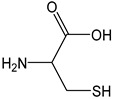	48	121.15	5.8	4.4	Eggs, broccoli,garlic, wheat germ
*Methionine*	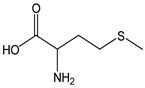	13	149.21	1.9	1.2	Eggs, sesame seeds, cashew nuts, fish, meat
*Istidine*	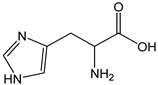	<1	155.16	-	<0.1	Beans, peanuts
*Hercynine*	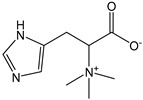	<1	197.23	-	<0.1	Mushrooms
*Ergothioneine*	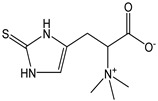	43	229.30	9.9	3.9	Mushrooms
*Glutathione*	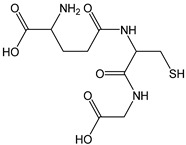	16	307.33	4.9	1.4	Spinach, tomato, pumpkin,strawberries
**Phenol compounds**						
*3,4-dihydroxy* *phenylacetic acid*	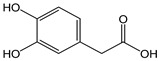	12	168.15	2.0	1.1	Olives, olive oil
*Vanillic acid*	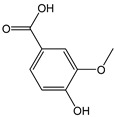	5	168.14	0.8	0.4	Coriander, onion,sage
*Gallic acid*	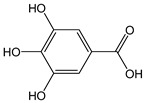	22	170.12	3.7	2	Raspberries, blackberries, strawberries, mangoes
*Syringic acid*	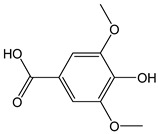	7	198.17	1.4	0.6	Walnuts, chard, chickpeas, peanuts, cocoa
*Ellagic acid*	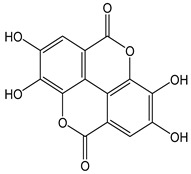	29	302.20	8.8	2.6	Blackberries, pomegranates, raspberries, strawberries, walnuts, grapes, goji berries
*Tannic acid*	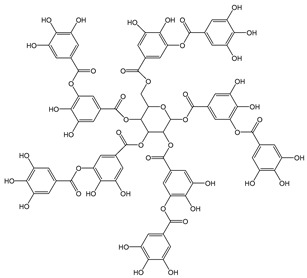	34	1701.19	57.8	3.1	Walnuts,green tea
*p-cumaric acid*	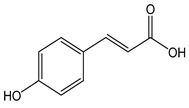	24	164.16	3.9	2.2	Chili, basil,walnuts, spinach,pineapple, thyme,sunflower
*Caffeic acid*	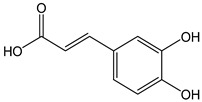	14	180.16	2.5	1.3	Coffee, chicory,peas, artichokes, strawberries
*Ferulic acid*	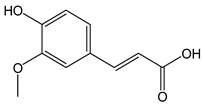	15	194.18	2.9	1.4	Oats, wheat, rice, apples, oranges, pineapple, coffee
*Curcumin*	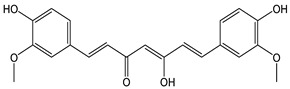	44	368.39	16.2	4.0	Turmeric
*Piceid*	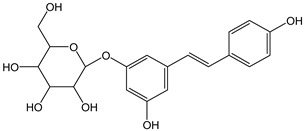	32	390.39	12.5	2.9	Grape juice
*Tyrosol*	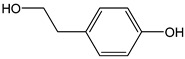	24	138.16	3.3	2.2	Olives, olive oil
*Hydroxytyrosol*	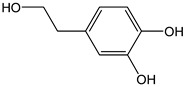	15	154.16	2.3	1.4	Olives, olive oil
*Oleuropein*	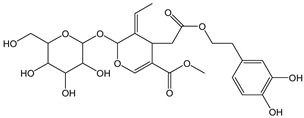	7	540.52	3.8	0.6	Olives, olive oil
*Chlorogenic acid*	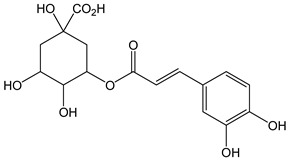	6	354.31	2.1	0.5	Coffee, apples,tomato, aubergine, cherries, pears, blueberries
*Cynarine*	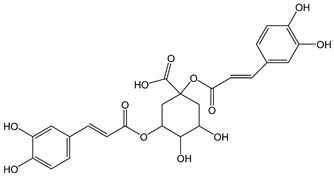	7	516.45	3.6	0.6	Artichokes
*Apigenin*	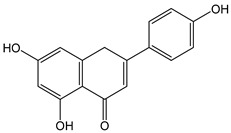	64	270.23	17.2	5.8	Chamomile, celery, parsley
*Kaempferol*	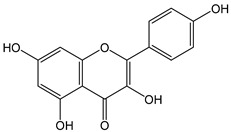	50	286.23	14.3	4.5	Apples, grapes, tomatoes, onions, potatoes, spinach, cucumbers, lettuce, peaches, blackberries
*Luteolin*	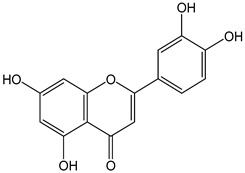	36	286.24	10.3	3.3	Carrots, peppers,olive oil, thyme peppermint, rosemary, oregano, lettuce, pomegranates, chocolate, capers, cucumbers
*Catechin*	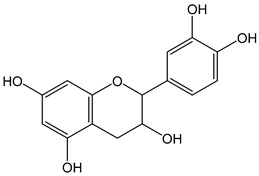	30	290.26	8.7	2.7	Green tea, cocoa,wine
*Quercetin*	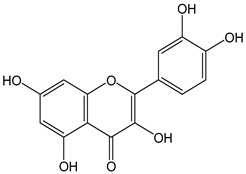	32	302.24	9.7	2.9	Capers, grapes, apples, red onion, celery
*Hesperetin*	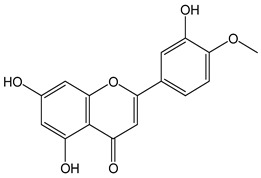	64	302.27	19.3	5.8	Citrus fruit, grapes
*Shaftoside*	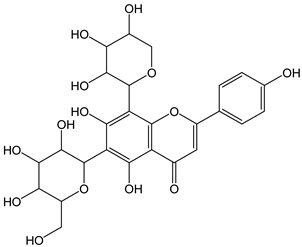	20	564.49	11.3	1.8	Carob germ, wheat germ
*Naringin*	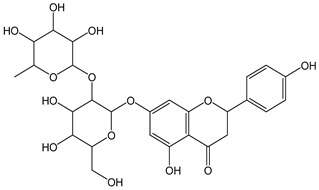	25	580.54	14.5	2.3	Grapefruit
*Rutin*	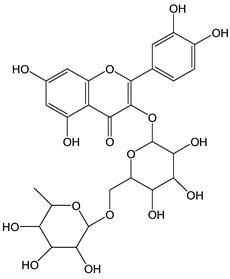	19	610.52	11.6	1.7	Citrus fruit, red wine, buckwheat
*Hesperidin*	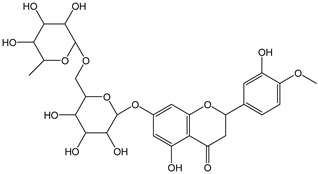	19	610.56	11.6	1.7	Citrus fruit
*Neohesperidin*	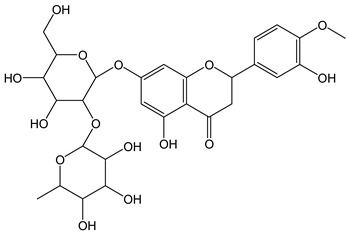	22	610.56	13.4	2.0	Citrus fruit
*Pelargonidin*	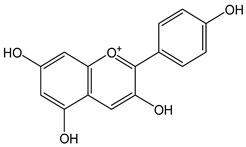	34	271.24	9.2	3.1	Raspberries, strawberries,blueberries, blackberries, pomegranates,plums,red beans
*Malvidin-3-O-galactosyde*	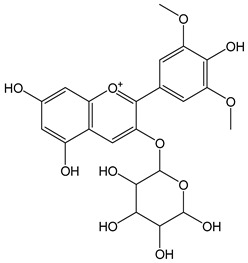	34	493.43	16.8	3.1	Blueberries
**Sterols**						
*Ergosterol*	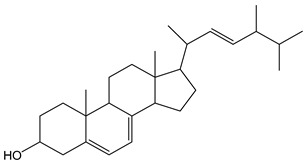	4	396.65	1.6	0.4	Mushrooms
**Terpenes**						
*Limonene*	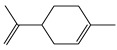	26	136.24	3.5	2.4	Citrus essential olis
*Linalool*	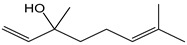	27	154.25	4.2	2.4	Essential oils
**Vitamins**						
*Nicotinic acid*	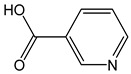	<1	123.11	-	<0.1	Cereals, meat, brewer’s yeast
*Ascorbic acid*	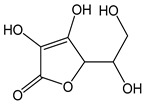	11	176.12	1.9	1	Citrus fruit, kiwis, tomatoes, peppers, cabbage
*Riboflavin*	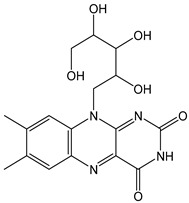	<1	376.36	-	<0.1	Milk, offal, kidney beans, egg whites
**Other**						
*Sulfite ion*	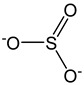	8	80.06	0.6	0.7	Food additive
*Furfural*	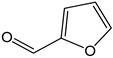	<1	96.08	-	<0.1	Neoformation product from cooking food
*Sorbate ion*	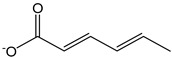	12	111.12	0.1	1.1	Food additive
*α-lipoic acid*	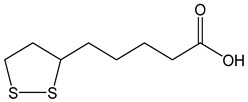	17	206.33	3.5	1.5	Dietary supplement
*Butylated hydroxytoluene (BHT)*	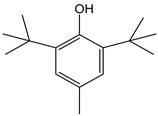	11	220.36	2.4	1	Food additive
*Trolox^®^*	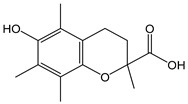	7	250.29	1.8	0.6	Reference standard

## Data Availability

The data that support the findings of this study are available on request from the corresponding authors.

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
