# Peer review of "Insights into the Structure–Capacity of Food Antioxidant Compounds Assessed Using Coulometry"

_antioxidants, 2023, doi:10.3390/antiox12111963_

Round 1
Reviewer 1 Report
Comments and Suggestions for Authors
The authors have investigated the structure-capacity of some food antioxidants using coulometry, which is interesting and meaningful for evaluating the antioxidant activity of food stuffs. The experiments are well designed and performed, and the results are also informative. The manuscript can be published after some minor issues being addressed:
1\ How did the authors define the “representative antioxidants”? Since some strong dietary antioxidants such as myricetin ,baicalein, and fisetin were not tested, what are the criteria for selecting detection objects?
2\ As I know, there are many literatures on the structure- activity relationships of dietary polyphenols, what’s the differences or new findings of the present study compared with the previous results.
3\There are some textual errors that need to be corrected:
Line14:delete “on”
Line36:“roles” to “role”
Line44:delete “it”
Line55:“undermining” to “undermining”
Line123:“boasts” to “boast”
Line191:“forms” to “form”
Line218:“is” to “are”
Line268:“defines” to “define”
Line332:delete “a”
Line375:“change” to “changes”
Line442-442:“demonstrating” to “demonstrated”
Line467:“underlie” to “underlies”
Comments on the Quality of English Language
minor revision
Author Response
Reviewer: The authors have investigated the structure-capacity of some food antioxidants using coulometry, which is interesting and meaningful for evaluating the antioxidant activity of food stuffs. The experiments are well designed and performed, and the results are also informative.
Authors: We gratefully thank the Reviewer who seized perfectly and appraised the meaning of our work.
The manuscript can be published after some minor issues being addressed:
R: 1\ How did the authors define the “representative antioxidants”? Since some strong dietary antioxidants such as myricetin, baicalein, and fisetin were not tested, what are the criteria for selecting detection objects?
A: This is a fair observation, because hundreds of food-derived constituents could be tested. We tried to assay representative components for the main classes of food antioxidants, especially based on their distribution and on availability. Of course, the study can be extended to other common food antioxidants. We included a related comment in the Experimental section (beginning of section 2.1).
R: 2\ As I know, there are many literatures on the structure- activity relationships of dietary polyphenols, what’s the differences or new findings of the present study compared with the previous results.
A: Many assays have been proposed for determining the antioxidant capacity of individual compounds or mixtures of compounds, often providing controversial data. As stated in the abstract and emphasized in the Conclusion, CDAC is an accurate, sensitive, rapid, and cheap measurement of the mol electrons (mol e-) transferred in a redox process and it is not related to reactions of antioxidants with specific probes. The mechanisms of antioxidant reactions vary with the chemical environment yielding results depending on the assay. Accordingly, what emerges from our research is that the mol e- transferred in a redox process can be even very different from those expected based on stoichiometry, because additional events may be involved in the oxidation mechanism. To meet the Reviewer’s suggestion, this concept has been further emphasized in the Conclusion section.
R: 3\There are some textual errors that need to be corrected:
Line14:delete “on”
A: The statement has been modified
R: Line36:“roles” to “role”
A: The point has not been changed because we intended that antioxidant can exert multiple different roles for food preservation. For instance, in addition to the antioxidant capacity, some of them well-described have antimicrobial properties.
R: Line44:delete “it”
A: the point has been corrected.
R: Line55:“undermining” to “undermining”
A: We are not sure to have right understood this suggestion. We have deleted “thus” before undermining
R: Line123:“boasts” to “boast”
A: we think that “boasts” is the right verbal form, because it is associated with CDAC (singular)
R: Line191:“forms” to “form”
A: the point has been corrected.
R: Line218:“is” to “are”
A: A typical plot [….] is…We think that “is” is the right verbal form
R: Line268:“defines” to “define”
A: The substitution defines. We think this is the right verbal form of “to define”
R: Line332:delete “a”
A: The point has been corrected
R: Line375:“change” to “changes”
A: The statement has been modified and clarified
R: Line442-442:“demonstrating” to “demonstrated”
A: Although we attempted to find it, we did not localize this point in the text. There is a “demonstrating at the line 424, but we think that is rightly placed.
R: Line467:“underlie” to “underlies”
A: Underlie is associated with “transfer processes” (Plural). So, the verbal form is correct.
Reviewer 2 Report
Comments and Suggestions for Authors
Coulometry is a powerful analytical technique that offers valuable insights into the intricate relationship between the chemical structure and the antioxidant capacity of various compounds found in our food. The assessment of food antioxidants using coulometry, often referred to as Coulometrically Determined Antioxidant Capacity (CDAC), has emerged as a method of choice for researchers seeking to unravel the mysteries of how these compounds function within the context of a complex food matrix.
This innovative approach involves the quantification of antioxidants present in individual compounds or their mixtures. It operates through constant-current coulometry, utilizing electrogenerated Br2 as the titrant, and employs bioamperometric detection to pinpoint the endpoint by measuring excess Br2. CDAC stands out for its accuracy, sensitivity, rapidity, and cost-efficiency in determining the number of electrons (mol e-) involved in redox processes.
A significant aspect of the research using CDAC is the investigation of the antioxidant capacity of a wide range of compounds commonly encountered in foods. These compounds vary greatly in their chemical structures, and understanding the factors that contribute to their antioxidant activity is crucial for both the food industry and nutrition science.
The results of CDAC studies have revealed intriguing findings. For instance, the molar ratio CDAC (CDAC) ranking of representative antioxidants often departs from what is traditionally expected based on structural motifs that promote electron or hydrogen atom transfers. This means that the mere chemical structure of an antioxidant does not solely determine its antioxidant capacity. These discoveries challenge established paradigms and stimulate further research into the nuanced mechanisms governing antioxidant behavior.
In addition to its role in unraveling the complex relationship between antioxidant compounds and their capacity to combat oxidative stress, CDAC provides a reliable method for estimating stoichiometric coefficients in the reactions of antioxidants with Br2. This contributes significantly to the understanding of the intricate chemistry at play in these reactions.
Moreover, CDAC research introduces the concept of the Electrochemical Ratio (ER), which compares the antioxidant capacity of individual compounds to the benchmark of ascorbic acid. ER offers a dimensionless nutritional index that enables a straightforward estimation of the cumulative antioxidant power of various foods. This, in turn, aids in evaluating the overall health benefits of different dietary choices.
In summary, Coulometry, as applied through CDAC, is a valuable tool for unlocking the secrets of food antioxidant compounds. It sheds light on the structure-activity relationships that underlie the (electro)chemical reactions these compounds undergo and provides a deeper understanding of their potential health benefits. This knowledge not only contributes to our understanding of nutrition but also informs the development of healthier and more effective food products, all of which is pivotal in our pursuit of better health and well-being.
You' should be right in noting that it's important for the experimental section of a research paper to clearly distinguish between the author's own work and data derived from existing literature or references. Here are some suggestions on how to improve the experimental section:
Clear Attribution: Ensure that it's explicitly clear which parts of the data and results are from your own research and which are based on prior literature. Clearly indicate sources for any data or methods you used that are not your own.
Detailed Descriptions: Provide a thorough description of the experimental procedures you followed, especially if they differ from established methods. This can help readers understand the novelty in your work.
Emphasize Novelty: If your experimental work contributes something new or innovative, be sure to highlight this in your paper. Explain how your approach or findings differ from existing research and what makes it significant.
Discussion of Success: After presenting the results, include a discussion section where you can emphasize the success and importance of your findings. Explain how your research advances the field, addresses a gap in knowledge, or provides new insights.
Comparative Analysis: If relevant, compare your results to existing literature or prior research. Discuss how your findings align or diverge from what has been reported before.
Reproducibility: Clearly state how others can replicate your experiments or methodology, and provide any necessary details, materials, or data to support this. This is crucial for scientific transparency.
Acknowledgment: If you received guidance, assistance, or equipment from others, make sure to acknowledge their contributions appropriately. This could include co-authors, collaborators, or institutions that provided support.
Before submitting your paper to a journal, consider having peers or mentors review it for clarity, accuracy, and completeness. They can provide valuable feedback on whether the experimental section effectively communicates the novelty and importance of your work.
Because these minor adjustments are suitable for publication in a journal.
Author Response
The Reviewer shows experience and expertise in the specific field of coulometry, and we are grateful to her/him for appraising our research. This Reviewer report appears as a non-conventional one, because it does not seem to recommend or suggest changes, but it rather indicates a series of general guidelines, which have in any case already been meet.
Based on these guidelines, we have slight modified the text in the “Results and Discussion” section to emphasize the current results and the novelty compared to the previously assessed knowledge.
Reviewer 3 Report
Comments and Suggestions for Authors
Report on the manuscript “Insights into the structure-capacity of food antioxidant compounds assessed by coulometry”, F. Siano et al.
Ref. Antioxidants-2688288.
General comments:
The manuscript describes the use of constant-current coulometry using 2 electrogenerated Br2 as the titrant, and bioamperometric detection of the end-point through Br2 excess to determine the antioxidant capacity of antioxidant components of foods. The manuscript presents an interesting idea, but involving several aspects requiring clarification. Accordingly, major revision is recommended based on the following considerations.
Remarks:
1) Several experimental details should be clarified. The authors indicate (lines 161-163): “Each CDAC measurement was performed with 0.1 mL of a methanol solution of individual molecules, accurately prepared at known concentration”. Apparently, the antioxidants were tested in MeOH solution. This offers an, in principle, generic problem: the estimated antioxidant capacity refers to conditions other than those operating in “biological” conditions (roughly, buffered aqueous solution).
2) Similarly, the authors declare in lines 160-161: “The glass vessel contained a M KBr solution; in the case of acidic analytes, it also contained 0.1 M H2SO4”. This indicates that the antioxidant capacities were estimated at different pH values. In principle, the logical practice would be to use a common buffered solution in order to compare the antioxidant capacities under the same conditions for all reagents.
3) This practice involves a conceptual problem. The authors indicate in the Introduction section, lines 125-127: “Thanks to the accurate determination of micromoles of electrons exchanged, the method is suitable for measuring the antioxidant capacity of poorly water-soluble compounds”. This means that the authors assume that the tested compounds have an intrinsic antioxidant capacity independent on the conditions where they act (i.e., independent on the solvent, pH, etc.).This is in contradiction with the previous text in lines 41-45: “According to a commonly accepted comprehensive definition, “antioxidant is any substance that delays, prevents or removes oxidative damage to a target molecule” [8]. In this sense, the antioxidant activity is related to the conditions of the as says through which it is evaluated, while the concept of antioxidant species might lose any biological significance outside the experimental environment [9]”.
4) Given the variable experimental conditions, it is not surprising that the rank of antioxidant capabilities obtained in the manuscript was divergent relative to the Bors criteria. This problem should be treated by the authors.
Author Response
R: The manuscript describes the use of constant-current coulometry using 2 electrogenerated Br2 as the titrant, and bioamperometric detection of the end-point through Br2 excess to determine the antioxidant capacity of antioxidant components of foods. The manuscript presents an interesting idea, but involving several aspects requiring clarification. Accordingly, major revision is recommended based on the following considerations.
A: We thank the Reviewer who appraised the rationale of our work.
R: Remarks: 1) Several experimental details should be clarified. The authors indicate (lines 161-163): “Each CDAC measurement was performed with 0.1 mL of a methanol solution of individual molecules, accurately prepared at known concentration”. Apparently, the antioxidants were tested in MeOH solution. This offers an, in principle, generic problem: the estimated antioxidant capacity refers to conditions other than those operating in “biological” conditions (roughly, buffered aqueous solution).
A: The Reviewer is right: the antioxidants were dissolved in methanol. Methanol was used only as a “vehicle” of the molecules and before measurements they get dissolved in a very large volume of an aqueous medium. We agree with the Reviewer that these are non-physiological conditions, as underlined in many points of the text, but our aim was to measure an intrinsic capability of antioxidant species to participate in redox events.
R: Similarly, the authors declare in lines 160-161: “The glass vessel contained a M KBr solution; in the case of acidic analytes, it also contained 0.1 M H2SO4”. This indicates that the antioxidant capacities were estimated at different pH values. In principle, the logical practice would be to use a common buffered solution in order to compare the antioxidant capacities under the same conditions for all reagents.
A: acidic pH was chosen only for assaying molecules containing the carboxylic functions in order to simplify the oxidation mechanisms and to prevent kinetics effects which become appreciably significant at neutral pH. Such effects, which are detectable in in vitro tests, does not occur in physiological environments owing to the presence of enzymes catalyzing redox processes. A systematic study of the effects of pH on the antioxidant capacity certainly is a very interesting perspective field of investigation as well as the determination of the in vitro antioxidant capacity of mixtures containing two or more antioxidants at varying concentrations.
R: This practice involves a conceptual problem. The authors indicate in the Introduction section, lines 125-127: “Thanks to the accurate determination of micromoles of electrons exchanged, the method is suitable for measuring the antioxidant capacity of poorly water-soluble compounds”. This means that the authors assume that the tested compounds have an intrinsic antioxidant capacity independent on the conditions where they act (i.e., independent on the solvent, pH, etc.). This is in contradiction with the previous text in lines 41-45: “According to a commonly accepted comprehensive definition, “antioxidant is any substance that delays, prevents or removes oxidative damage to a target molecule” [8]. In this sense, the antioxidant activity is related to the conditions of the assays through which it is evaluated, while the concept of antioxidant species might lose any biological significance outside the experimental environment [9]”.
A: Differently from the most common antioxidant capacity assays, the CDAC is a measurement of the mol electrons (mol e-) transferred in a redox process and it is not related to reactions of antioxidants with specific probes. The intrinsic reducing capacity of a species could represent an optimal reference also to evaluate the antioxidant effects also in physiologically relevant (bio)chemical environments. Please, also note that in this work we determined the antioxidant capacity which does not depends on additional kinetics parameters as the antioxidant activity does (ref. #15 of the revised version). We acknowledge that our concept was misleading, and we thank the Reviewer particularly for this comment. To clarify this point, a statemnt present in the Results of the original version has been moved to the introduction (Line 43-45 of the revised version) and two relevant references have been moved accordingly.
R: Given the variable experimental conditions, it is not surprising that the rank of antioxidant capabilities obtained in the manuscript was divergent relative to the Bors criteria. This problem should be treated by the authors.
A: We agree with the Reviewer that the divergence of antioxidant capacity ranking with Bors criteria should not be surprising, because the antioxidant capacity varies with the testing conditions. We included a statement and an example in the text (Section 3.2) to underline this aspect.
Reviewer 4 Report
Comments and Suggestions for Authors
In this manuscript, “Insights into the structure-capacity of food antioxidant compounds assessed by coulometry” by Siano et al. measures the CDAC of 48 individual antioxidants commonly 5 found in foods. Although, 48 individual antioxidants were measured, this work seems premature for publication. Therefore, I would suggest author may take at least a major revision. Here are the comments and suggestions:
1. Some introduction was duplicated from authors’ previous work in ref. #22.
2. Standard deviations should be added to the results throughout this work.
3. Authors are suggested to add some more figures.
4. Some in vitro or in vivo results can be collected and compared.
Author Response
Reviewer: In this manuscript, “Insights into the structure-capacity of food antioxidant compounds assessed by coulometry” by Siano et al. measures the CDAC of 48 individual antioxidants commonly 5 found in foods. Although, 48 individual antioxidants were measured, this work seems premature for publication. Therefore, I would suggest author may take at least a major revision. Here are the comments and suggestions.
Authors: The manuscript has been carefully revised based on the suggestions/recommendations of all the Reviewers. We hope that main concerns have been addressed and the manuscript can be suitable for publications.
R: 1. Some introduction was duplicated from authors’ previous work in ref. #22.
A: The procedure for CDAC was already described in ref. #22 (it is ref. #23 of the revised version), when it was applied to the determination of phenolic compounds in extra virgin olive oil samples. Since some steps of the procedure as well as some calculations needed modifications to meet the requirement of the new context (individual molecules, mostly water soluble, different concentrations), for the sake of clearness we preferred to describe again the determination of CDAC rather than detailing the differences compared to ref. #23. Please, also note that the device for measuring CDAC has changed, since in this work we used a purpose-made prototype electrochemical device integrating both the coulometric circuit and the biamperometric detection system, instead of a multi-necks glass vessel equipped with separate coulometric and biamperometric detection circuits. This last detail has been emphasized in the revised version.
R: 2. Standard deviations should be added to the results throughout this work.
A: The accuracy and repeatability of the assay are very high, and the relative standard deviation is in all cases (much) less than 2%. Therefore, in order not to burden the readability of Table 1, which is already large, we preferred to omit the standard deviation values. However, it was indicated in the table caption that the coefficient of variability of the measurements was in all cases less than 2%.
R: 3. Authors are suggested to add some more figures.
A: The manuscript contains two figures and many chemical structures are already depicted in Table 1. We could describe only some chemical processes in possible new figures, but we believe that this would lengthen the manuscript without improving its general clearness. For this reason, after carefully evaluating the Reviewer’s suggestion, we opted not to include new figures.
R: 4. Some in vitro or in vivo results can be collected and compared.
A: As the Reviewer knows, the in vitro and in vivo conditions related to the antioxidant capacity are very different and any comparison could be misleading. We purposely avoided correlating our in vitro data with the (few) in vivo data, both because the in vivo data are very controversial and few reliable data exist for the effects of acute administration on antioxidants, and because in the vast majority of cases foods, phytocomplexes or extracts containing numerous molecules rather than individual compounds were tested in vivo. Effects of antioxidants in cells are also controversial and often scarcely informative.
Please, also note that, as we stated in the Conclusion: “The indiscriminate use of antioxidant capacity has been rightly criticized since the health effects of antioxidants are not trivially correlated with their intake or with the concentration in foods or body fluids [62]”
Round 2
Reviewer 3 Report
Comments and Suggestions for Authors
The manuscript can be published in its current version
Author Response
We gratefully thank the Reviewer for endorsing the publication of our manuscript.
Reviewer 4 Report
Comments and Suggestions for Authors
Authors are suggested to plot some figures about the relation of functional groups with their measured antioxidant capacity with in-depth discussion.
Author Response
Gallic acid is a paradigmatic example of antioxidant that can lead to a multiplicity of oxidation reaction products. To comply with the fair Reviewer's suggestion, we introduced a new Figure (Figure 3 of the revised manuscript) schematizing the resonance in gallic acid semiquinone radical and evidencing one of the possible oxidative pathways, in particular the one leading to the formation of ellagic acid. We have also included a discussion of the figure and 2 new references.